# Real-World Data of a Group-Based Formula Low Energy Diet Programme in Achieving Type 2 Diabetes Remission and Weight Loss in an Ethnically Diverse Population in the UK: A Service Evaluation

**DOI:** 10.3390/nu14153146

**Published:** 2022-07-30

**Authors:** Owen Marples, Laura Resca, Julija Plavska, Samina Hassan, Vibhuti Mistry, Ritwika Mallik, Adrian Brown

**Affiliations:** 1Hackney Diabetes Centre, Homerton Healthcare NHS Foundation Trust, London E9 6SR, UK; laura.resca@nhs.net (L.R.); julija.plavska@nhs.net (J.P.); samina.hassan@nhs.net (S.H.); v.mistry1@nhs.net (V.M.); 2Endocrinology Department, Barts Health NHS Trust, London E1 1BB, UK; ritwika.mallik@nhs.net; 3Centre for Obesity Research, University College London, London WC1E 6JF, UK; 4Bariatric Centre for Weight Management and Metabolic Surgery, University College London Hospital NHS Trust, London NW1 2PG, UK; 5UCLH Biomedical Research Centre, National Institute of Health Research, London W1T 7HA, UK

**Keywords:** type 2 diabetes, diabetes remission, low energy diets, obesity, ethnic minority groups

## Abstract

(1) Background: Formula low energy diets (LED) are effective at inducing weight loss and type 2 diabetes (T2DM) remission. However, the effect of LED programmes in ethnic minority groups in the UK is unknown. (2) Methods: A service-evaluation was undertaken of a group-based LED, total diet replacement (TDR) programme in London, UK. The programme included: a 12-week TDR phase, 9-week food reintroduction and a 31-week weight maintenance phase and was delivered by a diabetes multi-disciplinary team. (3) Results: Between November 2018 and March 2020, 216 individuals were referred, 37 commenced the programme, with 29 completing (78%). The majority were of Black British (20%) ethnicity with a mean (SD) age of 50.4 (10.5) years, a body mass index of 34.4 (4.4) kg/m^2^ and a T2DM duration of 4.2 (3.6) years. At 12 months, 65.7% achieved T2DM remission, with a mean bodyweight loss of 11.6 (8.9) kg. Completers lost 15.8 (5.3) kg, with 31.4% of participants achieving ≥15 kg weight loss. Quality of life measures showed significant improvements. (4) Conclusions: This service evaluation shows for the first time in the UK that a group-based formula LED programme can be effective in achieving T2DM remission and weight loss in an ethnical diverse population.

## 1. Introduction

Type 2 diabetes (T2DM) continues to represent a significant and growing challenge for healthcare services [1,2]. People living with T2DM are at an increased risk of mortality and morbidity, with those from ethnic minority groups at particular risk [3] and the COVID-19 pandemic further increased vulnerability [4].

In the London Borough of Hackney (UK), 40% of residents identified as being from ethnic minority groups [5], with a proportionally higher prevalence of T2DM in Black and Asian groups [6]. Primary care data from the UK has shown Asian and Black groups have a greater likelihood of being diagnosed with T2DM [7]. It is recommended that people living with T2DM, and obesity should aim to lose approximately 15 kg following T2DM diagnosis in order to achieve T2DM remission [8]. However, clinical trials in people living with obesity and T2DM typically achieve weight losses of approximately 3.2 kg, with only a 1.46% cumulative prevalence of remission being reported in clinical practice [9]. Furthermore, while bariatric surgery helps individuals achieve significant weight loss and diabetes remission, due to its limited availability in the UK, possible associated complications and not all patients wishing to have surgery, it may not be appropriate or accessible in all cases [10]

The Diabetes Remission Clinical Trial (DiRECT) which used a LED programme resulted in 46% and 36% achieving T2DM remission at 1 and 2 years, respectively [11,12]. However, 98% were of White ethnicity, which raises the question if these results can be generalised, and specifically, if similar results are possible in a more ethnically diverse populations found in the UK. A recent study in Qatar which recruited Middle Eastern and North African ethnic groups found these programmes were effective, with 61% of participants with T2DM achieving remission at 12 months [13]; however, to date there is a paucity of data in the UK.

Group-based T2DM programmes have been suggested to provide superior clinical outcomes compared to one-to-one care [14]. With previous studies on T2DM remission delivered one-to-one in those diagnosed for less than 6 years [11,13], questions remain regarding the efficacy of group-based LED programmes, particularly in people from ethnic minority groups living with T2DM and those with longer disease duration. The aim of this service evaluation was to examine the impact of a group-based LED programme in an ethnically diverse population living with T2DM in the London borough of Hackney.

## 2. Materials and Methods

The Hackney Diabetes Centre (HDC) at Homerton University Hospital NHS Foundation Trust (HUH) was commissioned by the City and Hackney Clinical Commissioning Group (CCG) to investigate the feasibility and acceptability of a group-based formula LED programme in adults with T2DM in an ethnically diverse population.

### 2.1. Practitioner Training

The formula LED programme used was the Counterweight-Plus programme [15]. This programme was used in DiRECT [12] and implemented by a multidisciplinary team (MDT) at HDC, who had received Counterweight-Plus training and certification. The MDT consisted of two diabetes specialist dietitians (DSD), two diabetes specialist nurses (DSN) and one diabetes specialist psychological therapist. Counterweight Ltd. trainers provided additional supervision and mentoring throughout the programme.

### 2.2. Participant Population, Screening and Referral

Eligible participants were aged 18 to 75 years, with a confirmed diagnosis of T2DM (via Egton Medical Information Systems medical records), diagnosed with a glycated haemoglobin (HbA1c) >48 mmol/mol). A body mass index (BMI) equal or greater than 27 kg/m^2^ (if Asian equal or greater than 26 kg/m^2^) was required. Please see the Appendix A for the exclusion criteria.

### 2.3. Recruitment

The HDC conducted initiatives to increase awareness and referrals to the LED programme from primary and secondary care. These included HDC clinicians who discussed T2DM remission and the programme with eligible individuals in primary and secondary care clinics. A local referral form was developed, which aimed to facilitate a collaborative discussion between clinician and individuals regarding T2DM remission and programme expectations. Hackney CCG shared programme updates, outcomes and the referral form via their newsletter and website. Throughout the programme implementation outcomes were shared with Hackney CCG to help promote referrals and address clinician concerns [16].

Following referral, participants were screened by the HDC MDT, with eligible individuals being invited to a group “taster session”. This session provided further details regarding the LED programme, weight loss expectations, potential benefits, and side effects, with samples of the formula LED product being given to try or take home. Participants were also informed that the product cost a total of GBP 40 a week and they were expected to self-fund.

Individuals who wished to proceed were asked to complete a readiness to change questionnaire which asked them to rate their motivation and confidence on a scale (0–10, 0: not confident at all, 10: extremely confident) [17]. If an individual scored below 7 on either scale, they were not able to proceed without further self-reflection and consideration of how to increase their confidence and motivation [17]. All participants provided written informed consent and signed a form outlining their commitment to the programme, including session attendance. As this was a service evaluation, no formal ethical approval was required.

### 2.4. LED Programme

Participants were supported in groups by the trained HDC MDT. Sessions were conducted in a group-based face-to-face setting; however, during the COVID-19 pandemic and subsequent lockdowns, several sessions were run via video consultation. More information on how the sessions were run and adapted to the diverse community of Hackney can be found in the Appendix A. Participants were divided into four groups and each group started independently between July 2019 and January 2020 and were staggered to enable HDC to run the programme within staffing capacity.

### 2.5. LED TDR Phase (12 Weeks)

Participants followed a formula LED TDR (Counterweight TDR products ) with the addition of 50 mL low-fat milk for hot beverages which provided a total of 825–853 kcal/day (3452–3569 kJ/day) and a macronutrient breakdown of 61% carbohydrate, 13% fat and 26% protein. In total, four products were taken in the form of soups and shakes. Participants were also instructed to take a daily laxative twice a day (Fybogel) in addition to 2.25 litres of zero calorie fluid to proactively reduce constipation risk. Over the 12-week TDR phase, participants attended seven structured group sessions (bi-weekly). Content included goal setting, planning and side-effects and relapse-management, which focused on behaviour change techniques to facilitate long-term change [17].

### 2.6. Food Reintroduction Phase (9 Weeks)

Participants were instructed on the gradual reduction in the formula product and the incorporation of nutritionally dense and energy-restricted meals (360–400 calories per meal). A free copy of Carbs & Cals Very Low Calorie Recipes and Meal Plans [18] was provided to each participant to help with meal choices. Individual flexibility was provided regarding the rate at which meals were reintroduced, as well as participants’ preference for their long-term dietary approach, with guidance being provided for both low-fat and low-carbohydrate diets. This phase consisted of five group sessions (bi-weekly). Content included behavioural strategies, nutrition education, the encouragement of a structured meal pattern and lifestyle planning [17].

### 2.7. Weight Loss Maintenance (31 Weeks)

The weight loss maintenance (WM) phase consisted of eight sessions (monthly) which aimed to help participants maintain their weight loss while also limiting weight regain to no more than 2 kg by the end of the 12-month programme. If participants regained more than 2 kg they were offered a 4-week rescue plan of TDR or FR, depending on participant preference, goals and weight regain [17]. A bi-weekly review was then implemented, typically for a total of four weeks, by the DSD while liaising with the MDT accordingly. Sessions focused on relapse management, nutrition education, managing difficult circumstances (e.g., eating out) and behavioural strategies [17].

### 2.8. Rescue Plan

If participants regained more than 2 kg they were offered a rescue plan of TDR or FR depending on participant preference, goals and weight regain [17]. A bi-weekly review was then implemented, typically for a total of 4 weeks, by the DSD while liaising with the MDT accordingly.

### 2.9. Physical Activity Recommendations

During the LED TDR phase, participants were advised not to perform any high intensity exercise. At the FR phase, an increase in physical activity was encouraged via session content and individual goal setting—with the aim of at least 150 min of moderate activity a week [17]. This target was also encouraged throughout the WM phase. Exercise on referral was also offered from the FR phase, an initiative allowing clinicians to refer patients to a supervised exercise programme held by local services.

### 2.10. Medical Supervision and Psychological Support

Medication titration and cessation was overseen by two trained DSNs. All oral diabetes medications were stopped at baseline and insulin doses were titrated as per the Counterweight-Plus protocol [17]. Counterweight Ltd. provided supervision for practitioner queries regarding aspects of the programme and participant support. Practitioners also had access to a ‘frequently answered questions’ document, which provided guidance and rationales for aspects of medical management. Participants were offered up to five one-to-one psychological support sessions (45 min per session) provided by the psychological therapist.

### 2.11. Outcomes

The primary outcome was weight change at 12 months following LED programme commencement. The percentage of participants who achieved >5, 10, 15 and 20 kg of weight loss was also included. Secondary outcomes reported were improvement to HbA1c and diabetes remission rates, alongside other health-related outcomes including body composition, medication changes, blood pressure, blood lipids and mental health and eating behaviour at 12 months. Remission was defined as glycated HbA1c of less than 6.5% (<48 mmol/mol) after at least 2 months off all antidiabetic medications, from baseline to 12 months [12].

### 2.12. Data Collection

During group sessions, attendance, participant bodyweight and random plasma glucose were collected. Additionally, participants on blood pressure (BP) medication had their BP monitored on a session-by-session basis to allow effective medication adjustment.

Due to the COVID-19 pandemic, there were some difficulties attaining anthropometric and biochemical measurements at various stages throughout the programme. Following previous pragmatic processes [15], missing 12 month data, including participants who had withdrawn, were collected within 9–18 months of the participants’ expected programme completion date. Available, missing data were collected from primary care electronic patient records.

Patient mental health, quality of life and function in addition to binge eating behaviour were collected using the following validated questionnaires: Diabetes Distress Scale (DDS), Patient Health Questionnaire (PHQ9) [19], Generalised Anxiety Disorder (GAD-7) [20], Work and Social Adjustment Scale (WAS) [21], Rosenberg self-esteem scale [22] and Binge Eating Scale (BES) [23]. These were completed by the participants at the start of the programme and the end of LED TDR, FR and WM phases. Those who withdrew were asked to complete DDS, PHQ9, GAD-7, WAS and BES questionnaires at 12 months. Further detail regarding data collection can be found in Appendix A.

### 2.13. Statistical Analysis

Demographic data were summarised using mean (standard (SD)) for parametric data and median (interquartile (IQR)) for non-parametric data. Continuous data with categorical data were summarised using counts (percentages). To assess the differences in pre- and post-intervention, parametric and non-parametric data were analysed using appropriate tests (t-test; paired t-tests; Mann Whitney-U test) with Bonferroni correction used for multiple comparisons. Categorical data were analysed using chi-squared tests. Binomial logistic regression was used to assess predictors of participating in the LED programme and T2DM remission. For the LED programme participation sex, ethnicity, referral source, age, BMI and HbA1c were used as control variables. For the T2DM remission regression, weight loss at 3 and 12 months were used as control variables. These were added independently and additional factors of sex, number of diabetes medication were included. Odd ratios, 95% confidence intervals (CI) and *p*-values were reported for significant outcomes. The assumptions of each model were checked and met. Statistical analysis was conducted via SPSS Statistics (IBM, Version 26.0), and statistical significance was defined as a *p*-value < 0.05. Due to the fact that this was a service evaluation, formal sample size calculation was not conducted.

### 2.14. Ethics

This service evaluation of care required no formal ethical approval.

## 3. Results

### 3.1. Referrals and Participants

From November 2018 to March 2020, 216 individuals were referred to the LED programme, with 195 deemed eligible for a screening questionnaire (90%). Of these individuals 76 (39%) were invited to a taster session, with 37 (48%) of consenting to participate in the programme. Two participants were excluded from the analysis due to not having T2DM, in total 35 participants in total in the final analysis (Figure 1). Median attendance for completers was 17 sessions out of 20, with 29 participants completing the entire programme at 12 months. Six participants withdrew (17.1%), of these 66.7% (*n* = 4) and 33.3% (*n* = 2) withdrew in the TDR and WM phases, respectively. A total of 10 of the 35 participants (40%) accessed one-to-one psychological therapist support.

Of those referred to the programme (*n* = 216), the mean age was 52.5 (SD 11.0) years, the BMI was 35.7 (SD 6.5) kg/m^2^, HbA1c was 66.2 (SD 21.1) mmol/mol and 55.1% (*n* = 119) were women, with the majority of African (incl. Somali) ethnicity (*n* = 41, 19%). The mean length of T2DM diagnosis was 3.7 (SD 5.2) years. Please see Appendix A for further information regarding referral data. The regression analysis revealed that older participants were less likely to participate in the LED programmes (OR 0.94, 95% CI 0.95–0.99; *p* = 0.015). When comparing those that participated and those who did not, those who did not participate had higher depression scores (8.9 vs. 5.4, *p* = 0.024).

Of those participating in the programme (*n* = 35), the mean age was 50.4 (SD 10.5) years, BMI was 34.4 (SD 4.4) kg/m^2^, HbA1c was 65 (SD 20.3) mmol/mol (8%, SD 1.9) and 20 (57%) were men. The majority of participants were from either Black (*n* = 7, 20%) or White British (*n* = 7, 20%) ethnic groups and the mean length of diagnosis was 4.2 (3.6) years (Table 1).

### 3.2. Clinical Outcomes

T2DM remission was achieved in 65.7% (*n* = 23) at 12 months. Remission rates varied according to ethnic groups (Appendix A); however, this was not statistically significant. The mean reduction in HbA1c at 12 months was 15.5 mmol/mol (SD 19.5; *p* < 0.001) (1.4%, SD 1.8) (Figure 2). At 12 months there was a significant reduction in oral anti-hyperglycaemic agent use (*p* < 0.001), with 29 participants (82.9%) taking no medication. When comparing those who achieved remission and those who did not, those who achieved remission were on less medication at baseline (1.6 vs. 0.8, *p* = 0.021), and achieved greater weight loss at 3 and 12 months (17.7 vs. 7.5 kg, *p* < 0.001; 15.4 vs. 4.1 kg, *p* < 0.001). Other baseline demographics such as quality of life and binge eating showed no difference between the groups.

Binomial logistic regression was conducted to assess the predictors of T2DM remission, while individually, both weight loss at 3 and 12 months predicted participants achieving remission (OR 4.51, 95% CI 1.01–20.1, *p* = 0.048; OR 1.37, 95% CI 1.09–1.72, *p* = 0.006, respectively). When controlling for other variables, no factors increased the likelihood of T2DM remission.

Mean weight loss by ITT analysis was 14.2 kg (SD 6.5, *p* < 0.001) and 11.6 kg (SD 8.9, *p* < 0.001) at 3 and 12 months, respectively, with a significant weight regain of 2.7 kg (4.4, *p* < 0.001) between 3 and 12 months (Figure 3A). At 12 months, 6 participants (17.1%) achieved greater than 20 kg weight loss, 11 (31.4%) participants achieved greater than 15 kg, with 19 (54.3%) and 28 (80%) losing over 10 and 5 kg, respectively (Figure 3B). The mean weight loss of completers was 15.8 kg (5.4, *p* < 0.001) and 13.5 (8.0, *p* < 0.001) kg at 3 and 12 months, respectively.

In addition, there were improvements in body composition with significant reductions in fat mass at both 3 and 12 months (9.3 kg, SD 6.3; 10.0 kg, SD 9.2 kg, both *p* < 0.001, respectively) and waist circumference at both 3 and 12 months (8.6 cm SD 8.7, 12.9 cm, SD 9.0, both *p* < 0.001, respectively). There was also a significant reduction in fat free mass at 3 months of 7.0 kg (SD 6.9; *p* < 0.001) and this reduced to less than half at 3.2 kg (SD 6.8, *p* = 0.01) at 12 months.

There were significant and clinically relevant reductions observed in both systolic (9.3 mmHg, SD 16.2; *p* = 0.002) and diastolic (9.7 mmHg, SD 14.6; *p* < 0.001) blood pressure. The proportion of participants not taking antihypertensive medication at baseline was 40.0% (*n* = 14); this significantly increased to 58.8% (*n* = 20) at 12 months. Mean total, low-density lipoprotein cholesterol (LDL-C) and high-density lipoprotein cholesterol (HDL-C) did not significantly change, although there was a significant reduction in triglycerides (mean reduction 0.66 mmol/L, *p* < 0.001) (Table 2). From baseline to 12 months, mean self-esteem increased by 2.7 (SD 3.9, *p* < 0.001) and diabetes distress and depression scores decreased by 0.5 (SD 0.8, *p* = 0.001) and 1.9 (SD 4.7, *p* = 0.021), respectively. Anxiety, impairment in functioning and binge eating risk scores showed no significant change. No serious adverse events were reported; however, side-effects included constipation, diarrhoea, nausea, fatigue and feeling cold.

## 4. Discussion

The results of this service evaluation show for the first time in the UK that a group-based LED programme is an acceptable and effective way of achieving significant weight loss and T2DM remission in an ethnic minority population. Approximately, two thirds of participants achieved T2DM remission at 12 months. Reductions in diabetes and anti-hypertensive medications were noted, as well as significant improvements to self-esteem, diabetes distress and depression. The findings suggest T2DM remission can be achieved in an ethnically diverse population which is at higher risk of T2DM [24] and associated co-morbidities and complications [25]. Furthermore, weight loss and T2DM remission outcomes were greater than those reported in individual face-to-face care [11,13,15], suggesting that using group-based LED programme may help to effectively meet the rising demand for adult T2DM remission services in the UK [26].

Previous research using a one-to-one LED remission programme in those predominantly from a White ethic background resulted in mean weight losses of around 10 kg at 12 months [12], while our programme achieved a marginally greater weight loss (11.6 kg). Furthermore, a higher proportion of our participants achieved over 15 kg weight loss (31%) at 12 months, compared to the DiRECT (24%) and DIADEM-I (21%) clinical trials [12,13], with 17.1% of participants achieving a weight loss of more than 20 kg.

Growing evidence suggests a weight loss of 15 kg or more is associated with the greatest chance of T2DM remission via reductions in excess pancreatic and liver fat leading to increased pancreatic beta-cell function and hepatic insulin sensitivity [27]. This greater degree of weight loss may explain why our participants showed higher rates of remission (65.7%) at 12 months when compared to other studies [12,13] with the independent logistic regression showing weight losses at 3 and 12 months as independent predictors. When we controlled for other potential predictors, this was no longer significant. This may have been due to the limited numbers within the service evaluation and will require further investigation.

To our knowledge, this is the first service evaluation of a group-based LED programme in an ethnically diverse population in the UK and provides further evidence of the efficacy of LEDs for achieving T2DM remission in people living with T2DM. Ethnic minority groups in the UK have a higher risk of T2DM and have poorer health outcomes compared to people from a White background [7,25]. As a result, it is vital diabetes services provide culturally appropriate care tailored to the needs of their local community [16]. To achieve this, the HDC-adapted elements of the Counterweight-Plus programme to ensure culturally sensitive dietary advice were provided such as resources outlining nutritional advice centred on traditional, every-day foods. It was noted that 19% of those referred were of African (including Somali)background, yet of those who participated, only 11.4% represented this group. The reason for the low uptake may be related to several barriers to accessing healthcare and having to self-pay for the product [25].

People living with T2DM have an increased risk of having poorer mental health including diabetes distress, depression, and anxiety [28]. Importantly, the participants showed small though significant improvements in depression, diabetes distress and self-esteem. In addition, previous research has only focused on quality-of-life scores [11,12,13]; these are novel data showing that LED programmes help to improve other aspects of mental health related to T2DM. In addition, there was no change in binge eating disorder risk which is reassuring and supports evidence that formula LED do not increase binge eating risk in people living with obesity [29]. When looking at surrogate markers of cardiovascular disease (CVD), as seen in previous studies [11,12,13], there was a significant reduction in triglycerides but no change in other lipid markers. This might be explained by cholesterol being within target ranges at baseline and that lipid-lowering drugs were being taken and these continued during the evaluation. Importantly, blood pressure was significantly reduced in our participants, greater than has been seen in previous studies, while in parallel blood pressure medication was significantly lowered, with nearly 60% being on no blood pressure medication, and having normal blood pressure. Overall, these beneficial effects show a reduction in cardiometabolic risk in an ethnically diverse population traditionally with a higher risk of CVD and mortality [30].

Despite our programme utilising a similar LED programme to previous research [11,13,15], there were several important differences. The programme was delivered in groups and used both face-to-face and remote formats, suggesting that this may be a useful way to increase access to remission services in traditionally hard to reach populations. Our participants were predominately from ethnic minority groups, and as previously mentioned were in contrast to DiRECT [12] that involved 98% White participants. Similar to DiRECT [12] and DIADEM-I [13], our programme had a larger proportion of men but a longer duration of T2DM (4.2 years) [12,13]. Additionally, 10 participants had a diagnosis duration of over six years, 4 of which achieved remission at 12 months. This may have been due to 77.2% of the participants being on only one at baseline, which has been shown to be a predictor of achieving T2DM remission [31]. This would suggest that even patients with long-standing T2DM may benefit from being referred to LED programmes, as long as they are not on multiple antidiabetic medications.

The DiRECT study had a similar percentage of participants treated by diet alone at baseline (24.2%) [12], while DIADEM-I was lower (10.2%) [13]. This may be reflective of a similar pharmacotherapy practice in the UK compared to the Middle East [13].

To date there remains limited published data from clinical services in the UK using LED programmes on people living with obesity or T2DM. In those studies that published their service evaluation data using a LED programme (*n* = 288, 14.2 kg) in people living with obesity [15], the programme led to clinically significant weight loss (*n* = 88, 14.2 kg). Interestingly, our programme achieved a greater amount of weight loss (15.8 kg) at 12 months, despite our programme containing a participant group with a significantly lower mean BMI (34.4 kg/m^2^ vs. 45.7 kg/m^2^) and a greater proportion living with T2DM (100% vs. 34.4%) [15], both of which have traditionally been indicators of poorer weight loss [32,33]. This would suggest that the participants were able to maintain dietary adherence despite using a group-based programme, participants self-funding their meal replacement products and several groups running over a global pandemic. It is not entirely clear why this was the case, although a recent meta-analysis suggests group-based weight loss interventions may provide superior weight loss outcomes compared to one-to-one care [34] and rapid and early weight loss may lead to higher participant motivation and improved weight maintenance [15,16]. Service provision by a specialist MDT was a unique feature of our programme and may have contributed to superior weight loss outcomes [35] compared to previous studies [12,15], alongside the cultural adaptions made to meet participants needs. As a result, the combination of group-based service provision by a specialist MDT utilising a LED programme may have helped offset the potential negative impact on participant attendance caused by self-funding the LED product [36,37]. The dropout rate (%) of our programme was slightly less than DiRECT [12] and DIADEM-I [13], suggesting that this model may be an acceptable means of delivery to people living with T2DM from ethnic minority groups. Due to the limited numbers of participants, further research is needed to confirm this finding.

There were several limitations to our service evaluation. The programme was conducted in a secondary care outpatient setting to facilitate MDT support and close monitoring. This is a care model which may not be as cost effective as primary care-based LED programmes [38]. Due to the COVID-19 pandemic, the service adapted to a digital healthcare model via group-based video conferencing and participants self-monitoring and reporting. This suggests that with the necessary adaptations to delivery and monitoring protocols, a LED programme could be transferred to the digital care setting, helping provide effective [39] and accessible [40] care. While physical activity was encouraged, especially following the LED TDR phase, these outcomes were not measured at any point during the programme. Further limitations lie in the small sample size and the fact that this was not a randomised control design; therefore, a cause—effect relationship cannot be confirmed.

## 5. Conclusions

This service evaluation provides evidence for the first time that a group-based LED remission service is acceptable and efficacious in an ethnically diverse population in London. Weight loss and remission rates at 12 months were greater than previous studies [12], despite containing a more ethnically diverse population with a longer duration of T2DM and a higher baseline HbA1c. This suggests that with culturally appropriate modifications to a LED programme, using a self-funding model, individuals from ethnic minority groups will both take part in and adhere to a LED T2DM remission programme.

## Figures and Tables

**Figure 1 nutrients-14-03146-f001:**
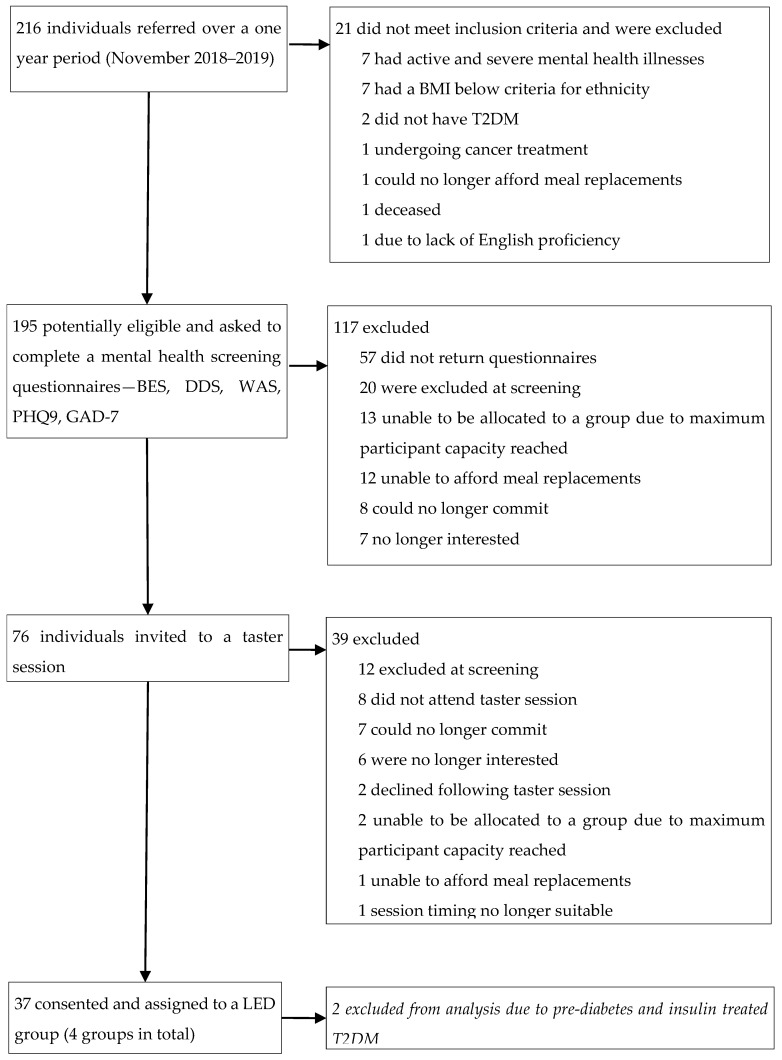
Trial profile. BMI, body mass index, T2DM; type 2 diabetes mellitus; DDS = Diabetes Distress Score. PHQ9 = Patient Health Questionnaire. GAD7 = Generalised Anxiety Disorder Questionnaire. WAS = Work and Social Adjustment Scale. BES = Binge Eating Scale.

**Figure 2 nutrients-14-03146-f002:**
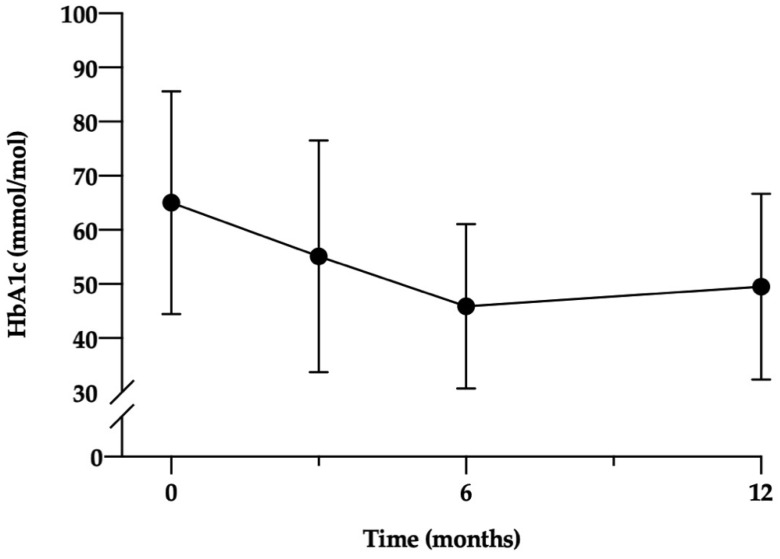
HbA1c outcome of participants over 12 months. HbA1c, glycated heamoglobin.

**Figure 3 nutrients-14-03146-f003:**
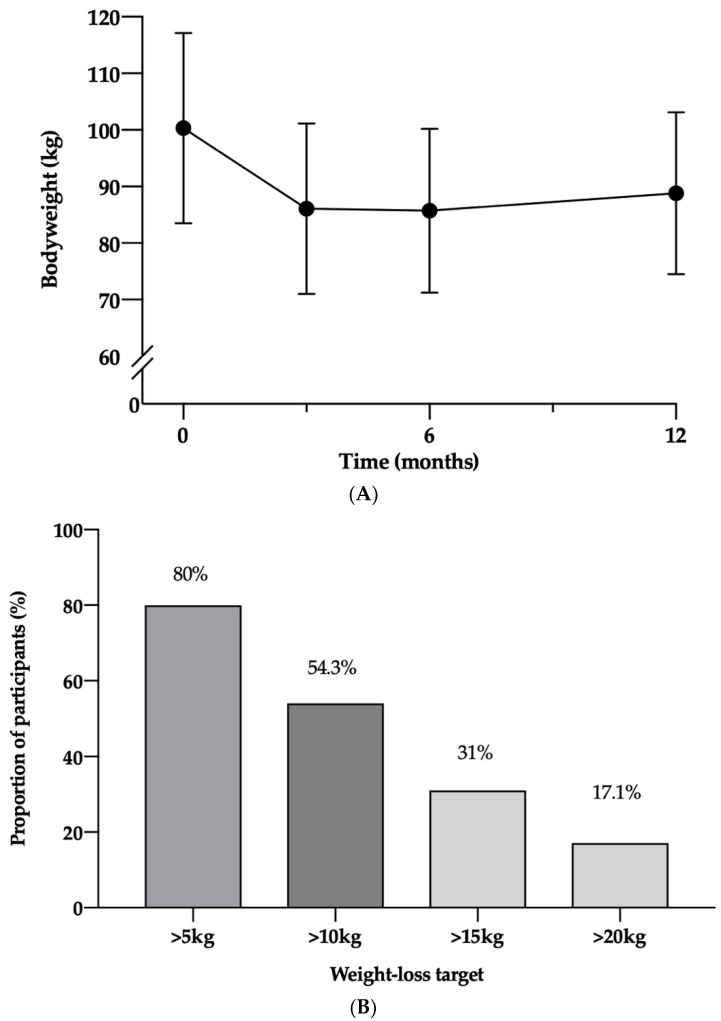
(**A**). Bodyweight of participants over 12 months (error bars represent 95% CIs) (**B**). Proportion of participants achieving key weight loss targets at 12 months. Kg, kilograms, %, percentage.

**Table 1 nutrients-14-03146-t001:** Baseline characteristics of participants.

Characteristics	Participants (*n* = 35)
Men, *n* (%)	20 (57)
Women, *n* (%)	15 (43)
Age, years (SD)	50.4 (10.5)
Ethnicity, *n* (%)	
Black British	7 (20%)
White British	7 (20%)
Caribbean	6 (17.1%)
African (incl. Somali)	4 (11.4%)
Other Black	3 (8.6%)
Pakistani (incl. British)	2 (5.7%)
Bangladeshi (incl. British)	2 (5.7%)
Any other mixed	2 (5.7%)
Irish	2 (5.7%)
Bodyweight, kg (SD)	100.3 (16.7)
BMI, kg/m^2^ (SD)	34.4 (4.4)
Waist circumference, cm (SD)	113.2 (11.3)
Body fat % (SD)	37.5 (8.6)
Duration of diabetes, years (SD)	4.2 (3.6)
Number of diabetes medications, *n* (%)	
0	8 (22.9)
1	19 (54.3)
2	6 (17.1)
≥3	2 (5.7)
Type of treatment / diabetes medication, *n* (%)	
Diet only	8 (22.9%)
Metformin	26 (74.3%)
Sulfonylurea	3 (8.6%)
SGLT2 inhibitor	2 (5.7%)
HbA1c, mmol/mol (SD)	65 (20.3)
HbA1c, % (SD)	8 (1.9)
Systolic blood pressure, mm Hg (SD)	133.6 (14.7)
Diastolic blood pressure, mm Hg (SD)	88.2 (9.5)
Hypertension, *n* (%)	18 (51.4%)
Number of antihypertensive medications, *n* (%)	
0	14 (40.0%)
1	13 (37.1%)
≥2	8 (22.9%)
Cardiovascular disease, *n* (%)	1 (3%)
Total cholesterol, mmol/L (SD)	4.8 (1.5)
HDL cholesterol, mmol/L (SD)	1.2 (0.4)
LDL cholesterol, mmol/L (SD)	2.6 (1.1)
Triglycerides, mmol/L (SD)	2.1 (2.1)
Psychological wellbeing and binge eating scores (SD)	
Rosenberg	21.3 (5.1)
DDS, mean	2.3 (0.9)
PHQ9	5.3 (4.7)
GAD7	3.7 (4.3)
WAS	6.3 (6.6)
BES	9.3 (5.8)

Data are in *n* (%), mean (SD). %, percentage; *n*, number; BMI, body mass index, HbA1c, glycated haemoglobin; kg/m^2^, kilograms per metre squared, kg, kilograms; DDS = Diabetes Distress Score. PHQ9 = Patient Health Questionnaire. GAD7 = Generalised Anxiety Disorder Questionnaire. WAS = Work and Social Adjustment Scale. BES = Binge Eating Scale; mmol/L, millimoles per litre. HDL, high-density lipoproteins, LDL, low-density lipoproteins; SGLT2, Sodium-glucose Cotransporter-2.

**Table 2 nutrients-14-03146-t002:** Summary of results in participants: key outcomes at 12 months.

			Mean (SD)			
	*n*	Baseline	12 Months	Change	95% CI	*p* Value
Weight (kg)	35	100.3 (16.8)	88.8 (14.3)	11.6 (8.9)	8.5–14.6	<0.001
BMI (kg/m^2^)	35	34.4 (4.5)	30.5 (4.2)	3.9 (2.8)	2.9–5.0	<0.001
Waist circumference (cm)	35	113.2 (11.5)	100.3 (10.8)	12.9 (8.9)	9.9–16.0	<0.001
Fat mass (%)	35	37.5 (8.8)	32.2 (9.6)	5.3 (4.9)	3.6–7.0	<0.001
Fat free mass (%)	35	62.5 (8.7)	67.8 (9.6)	−5.3 (5.0)	3.6–7.0	<0.001
HbA1c (mmol/mol)	35	65.0 (20.6)	49.5 (17.1)	15.5 (19.5)	8.8–22.2	<0.001
HbA1c (%)	35	8.1 (1.9)	6.7 (1.6)	1.4 (1.8)	0.8–2.0	<0.001
Number of diabetes medications (mean, SD)	35	1.1 (0.8)	0.3 (0.7)	0.8 (0.6)	0.6–1.0	<0.001
Systolic blood pressure (mm Hg)	35	133.6 (14.9)	124.2 (10.6)	9.3 (16.2)	3.8–14.9	0.002
Diastolic blood pressure (mm Hg)	35	88.2 (9.7)	78.5 (12.0)	9.7 (14.6)	4.7–14.7	<0.001
Number of anti-hypertensive medications	35	0.9 (1.0)	0.6 (0.2)	0.3 (0.5)	0.5–3.5	0.001
Total cholesterol (mmol/l)	35	4.8 (1.5)	4.5 (1.3)	0.3 (1.1)	−0.8–0.6	0.128
HDL cholesterol (mmol/l)	35	1.2 (0.4)	1.3 (0.4)	0.1 (0.2)	−1.3–0.1	0.74
LDL cholesterol (mmol/l)	34	2.7 (1.1)	2.6 (0.8)	0.1 (0.8)	−0.2–0.4	0.617
Triglycerides (mmol/l)	35	2.1 (2.2)	1.5 (1.8)	0.7 (0.9)	0.4–1.0	<0.001
Rosenberg	35	21.3 (5.2)	24.1 (5.0)	−2.7 (3.9)	−4.0–−1.4	<0.001
DDS Mean	35	2.3 (1.0)	1.8 (0.9)	0.5 (0.8)	0.2–0.7	0.001
PHQ9	35	5.3 (4.8)	3.4 (4.4)	1.9 (4.7)	0.3–3.5	0.021
GAD7	35	3.7 (4.4)	3.4 (4.7)	0.3 (3.7)	−0.9–1.6	0.584
WAS	35	6.3 (6.7)	6.5 (9.5)	−0.2 (8.2)	−3.0–2.6	0.869
BES	35	9.3 (6.0)	7.7 (5.7)	1.5 (6.1)	−0.6–3.6	0.144

Data are in *n* (%), mean (SD). %, percentage; *n*, number; BMI, body mass index, HbA1c, glycated haemoglobin; kg/m^2^, kilograms per metre squared, kg, kilograms; DDS = Diabetes Distress Score. PHQ9 = Patient Health Questionnaire. GAD7 = Generalised Anxiety Disorder Questionnaire. WAS = Work and Social Adjustment Scale. BES = Binge Eating Scale; mmol/L, millimoles per litre. HDL, high-density lipoproteins, LDL, low-density lipoproteins.

## Data Availability

Data available on request due to restrictions, e.g., privacy or ethical. The data presented in this study are available on request from the corresponding author. The data are not publicly available due to this involving NHS patients and not having consent to share on a public publicly accessible repository.

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
