# Peer review of "Real-World Data of a Group-Based Formula Low Energy Diet Programme in Achieving Type 2 Diabetes Remission and Weight Loss in an Ethnically Diverse Population in the UK: A Service Evaluation"

_nutrients, 2022, doi:10.3390/nu14153146_

Round 1

Reviewer 1 Report

The manuscript of Owen Marples and coll seems to be interesting, however major bias in the design of the study arose reading the manuscript and for this reason.

In particular, the aim of the study is not clear and well stated and the results presented are confusing due to the absence of a specific aim and give the reader a lot of information without a proper direction.

Further, if the aim of the manuscript would be to assess the efficacy of LED program in specific ethnical groups, a sample size evaluation should be carried out for the study design.

Author Response

Response to Reviewer: 1

Thank you very much for your comments, we have addressed your direct questions below and outlined associated changes to the manuscript. For clarity, we have listed the reviewer questions in bold below. To reflect the contributions of the Dr Ritwika Mallik, we have added her to the author list and contribution section.

  • The aim of the study is not clear and well stated and the results presented are confusing due the absence of specific aim. This gives the reader a lot of information without proper direction

Many thanks for this valuable comment and we apologise if the aims of our service evaluation were not clear and the impact this had on the interpretation of the results.

The aim of our service evaluation has now been clarified within the introduction as follows – “The aim of this service evaluation was to examine the real-world impact of a group-based LED programme in an ethnically diverse population living with T2DM in the London borough of Hackney” [Line 62–65].

As a result, the subsequent lines have been moved into the methods section – “The Hackney Diabetes Centre (HDC) at Homerton University Hospital NHS Foundation Trust (HUH) was commissioned by City & Hackney Clinical Commissioning Group (CCG) to investigate the feasibility and acceptability of a group-based formula LED programme in adults with T2DM in an ethnically diverse population” [Line 68–72].

Furthermore, we have changed the title to reflect the nature of the data presented by removed efficacy and replacing this with ‘real’, therefore it now reads – “Real-world data of a group-based formula low energy diet programme in achieving type 2 diabetes remission and weight loss in an ethnically diverse population in the UK: A service evaluation”.

To provide more clarity regarding the reported of the manuscript outcomes the following text has been added to the methods section – “The primary outcome was weight change at 12 months following LED programme commencement. The percentage of participants who achieve ≥5, ≥10, ≥15 and ≥20kg of weight loss was also included. Secondary outcomes reported were improvement to HbA1c and diabetes remission rates, alongside other health-related outcomes including body composition, medication changes, blood pressure, blood lipids, and mental health and eating behaviour at 12 months [Line 170–175].

The outcomes included within this service evaluation are reflective of previously published trials and a service evaluations in the area of LED interventions and T2DM remission. This was done in part to allow for comparison of outcomes between our data the that of others. These include:

  • Lean et al. Durability of a primary care-led weight-management intervention for remission of type 2 diabetes: 2-year results of the DiRECT open-label, cluster-randomised trial. Lancet Diabetes Endocrinol. 2019 May;7(5):344-355. doi: 10.1016/S2213-8587(19)30068-3.
  • Taheri, et al. Articles Effect of Intensive Lifestyle Intervention on Bodyweight and Glycaemia in Early Type 2 Diabetes (DIADEM-I): An Open-Label, Parallel-Group, Randomised Controlled Trial; 2020; Vol. 8).
  • McCombie, L. et al. Filling the Intervention Gap: Service Evaluation of an Intensive Nonsurgical Weight Management Programme for Severe and Complex Obesity. Journal of Human Nutrition and Dietetics 2019, 32, 329–337, doi:10.1111/jhn.12611.

Furthermore, to improve the flow of the results section for the reader. The following changes have been made:

  • Figure 3A & B title has been moved above the figures [Line 292–293].
  • To add clarity to the results section, we have added subtitles to the following sections ‘highlight referral and participant results’ (1), followed by ‘clinical outcomes’ (3.2).
  • We have removed some of the within text referral data, as much of this information is provided within supplementary material. Therefore, this now reads: “Of those referred to the programme (n=216), mean age was 52.5 (SD 11.0) years, BMI 35.7 (SD 6.5) kg/m2, HbA1c 66.2 (SD 21.1) mmol/mol, 55.1% (n=119) were women, with the majority being from African (incl. Somali) ethnicity (n=41, 19%). The mean length of T2DM diagnosis was 3.7 (SD 5.2) years. Please see Supplementary Material (Table S1) for further information regarding referral data to the programme” [Line 230 – 234].
  • Additionally, we did the same to the baseline characteristic data. This now reads: "Of those participating in the programme (n=35), mean age was 50.4 (SD 10.5) years, BMI was 34.4 (SD 4.4) kg/m2, HbA1c was 65 (SD 20.3) mmol/mol and 20 (57%) were men. The majority of participants were either Black (n=7, 20%) or White British (n=7, 20%) , and the mean length of diagnosis was 4.2 (3.6) years (Table 1)” [Line 238–241].
  • By reducing the above two paragraphs, less within text data is provided, but the reader is still able to compare key data between referrals and participants.

  • The aim of the manuscript would be to assess the efficacy of LED program in specific ethnic groups, a sample size evaluation should be carried out for the study design.

Thank you very much for this comment. We would like to clarify that this was a service evaluation and not a formal research study, therefore, a sample size evaluation was not carried out as part of the study design. To aid clarity within the manuscript we have added the following – “Due to this being a service evaluation, a formal sample size calculation was not conducted” [Line 210–211].

Further to the reviewer's helpful comments, please find below how we have addressed the areas within the review form that were identified as requiring improvement:

  • Does the introduction provide sufficient background and include all relevant references?
    • The authorship has reviewed the introduction to check for sufficient background and referencing and have made minor changes to aid clarity, please see submitted manuscript and comments below.
    • Each cited reference has been reviewed and the authorship feel they are relevant to the data presented and up to date.
    • Additional background has been provided regarding the importance of exploring the impact of real-world LED programmes in the UK – “Furthermore, while bariatric surgery helps individuals achieve significant weight loss and diabetes remission, due to its invasive nature and limited availability in the UK, it may not be appropriate or accessible in all cases” [Line 46–49].
    • An aim of the service evaluation has been clarified as stated above as follows - “The aim of this service evaluation was to examine the real-world impact of a group-based LED programme in an ethnically diverse population living with T2DM in the London borough of Hackney” [Line 62–65]. The title of our manuscript has also been changed to remove the word “efficacy” which may be a source of confusion to the reader and outside the scope of a service evaluation
  • Are all the cited references relevant to the research?
    • Each cited reference has been reviewed and the authorship feel they are relevant to the data presented and up to date.
  • Is the research design appropriate?
    • The rationale as to why no sample size calculation was provided has been clarified above in addition the following text has been added to the manuscript - “Due to this being a service evaluation, a formal sample size calculation was not conducted” [Line 210 – 211].
    • Primary and secondary outcomes have now been outlined within the methodology to help navigate the reader and provide clarity to the results - 11. Outcomes. The primary outcome was weight change at 12 months following LED programme commencement. The percentage of participants who achieve ≥5, ≥10, ≥15 and ≥20kg of weight loss was also included. Secondary outcomes reported were improvement to HbA1c and diabetes remission rates, alongside other health-related outcomes including body composition, medication changes, blood pressure, blood lipids, and mental health and eating behaviour at 12 months [Line 170 – 175].
  • The results are clearly presented?
    • As previous outlined, we have reduced the amount of within text data in the results section to reduce risk of confusion and aid clarity. We have retained some of the in text data between referrals and participants to help the reader compare both groups.
    • Figure 3A & B title has been moved above the Figures [Line 292–293].
    • To help navigate the reader through the results section, we have added subtitles to highlight referral and participant results (1) [Line 215], followed by clinical outcomes (3.2) [Lines 247].
  • Are the conclusions supported by the results?
    • We believe the conclusions are supported by the result. Due this being a service evaluation no firm conclusions can be reached regarding the efficacy of an LED programme. However, the results suggest this intervention can be beneficial when delivered in group format to people from an ethnically diverse population. This is important data, yet to be shown in the literature to date.

We would like to thank the reviewer again for their time and valuable comments and hope that this has clarified the comments and points they have regarding the manuscript.

Reviewer 2 Report

This is a meaningful study to demonstrate the usefulness of diabetes care in an ethnically diverse population. However, there are some points that need to be modified.

1.

In Table 2, why the P-value was not significant for changes in blood pressure?

Since the 95% CI was showed in the table, isn't it remove the P-value to avoid confusion?

2.

The impact of 10 kg weight loss on diabetes is significantly different between a 100 kg patient and a 50 kg patient.

Why authors are focusing on especially weight changes rather than BMI changes in this study?

3.

While the participants of this study include the elderly, were there any significantly differences between the elderly and the non-elderly in interpretation of the results?

For example, there are previous studies that do not recommend a reduction of BMI in the elderly.

4.

How much physical exercise the participants were already doing before the study baseline?

Author Response

Response to Reviewer: 2

Thank you very much for your comments, we have addressed your direct questions below and outlined associated changes to the manuscript. For clarity, we have listed the reviewer questions in bold below. To reflect the contributions of the Dr Ritwika Mallik, we have added her to the author list and contribution section.

  • In table 2, why the P-value was not significant for changes in blood pressure? Since the 95% CI was shown in the table, isn’t it remove the P-value to avoid confusion?

Thank you for noticing this. We have reviewed the statistical analysis and noticed an error in the reporting of the blood pressure P-values outlined in Table 2. This has been updated to reflect the correct P-values as follows: ‘SBP 0.002’ and ‘DBP <0.0001’ within Table 2. On discussion with the authorship we have chosen to kept both 95% CI and P-values within Table 2, unless the reviewer would prefer us to remove one, which we are happy to do. 

  • The impact of 10kg weight loss on diabetes is significant different between 100 kg patient and a 50 kg patient. Why are authors focusing on especially weight changes rather than BMI changes in this study?

We thank the reviewer for this comment. We agree with the reviews that there is s difference between weight loss of 10kg between a person who is 100kg and 50kg. The reason for us focusing on weight loss change rather than BMI change is that a substantial amount of published data has identified that greater weight loss, specifically ≥15kg, following the use of a LED intervention, is the primary predictor of T2DM remission. As a result, we have focused on weight loss and not BMI change, as this better reflects the current evidence base and allows for more direct comparison between our data and others.

The following studies and guidelines are provided to support this rationale:

  • Lean, et al. Durability of a Primary Care-Led Weight-Management Intervention for Remission of Type 2 Diabetes: 2-Year Results of the DiRECT Open-Label, Cluster-Randomised Trial. The Lancet Diabetes and Endocrinology 2019, 7, 344–355, doi:10.1016/S2213-8587(19)30068-3.
  • Thom G, et al. Predictors of type 2 diabetes remission in the Diabetes Remission Clinical Trial (DiRECT). Diabet Med. 2021 Aug;38(8):e14395. doi: 10.1111/dme.14395.
  • Evidence-based nutrition guidelines for the prevention and management of diabetes. 2018. Available at: https://www.diabetes.org.uk/professionals/position-statements-reports/food-nutrition-lifestyle/evidence-based-nutrition-guidelines-for-the-prevention-and-management-of-diabetes
  • Additionally, 10kg weight loss was a focus as DiRECT showed an average weight loss of 10kg at 12 months in the LED intervention arm.

  • While the participants of this study include elderly, were there any significant differences between the elderly and the non-elderly in interpretations of the results?

We thank the review for their comment. This is an important point and one we would have been keen to explore within our evaluation. As the reviewer will be aware the within medical research and ‘elderly’ person is defined as a person of 65 years of age or above, within this in mind we reviewed out data to see if such a comparison could be made. On review we identified that we only had three participants over this age, this therefore, precluded multi-variant analysis to assess the difference between elderly and non-elderly.

  • How much physical exercise were the participants already doing before the study baseline? 

Thank you for this question. Unfortunately, we did not measure participant baseline or follow-up physical activity levels. This is mentioned in the Discussion section which covers the limitations of our study – “While physical activity was encouraged, especially following the LED TDR phase, these outcomes were not measured at any point during the programme” [Line 441 - 443].

Further to the reviewer’s helpful comments, please find below how we have addressed the areas within the review form that were identified as requiring improvement:

  • Is the research design appropriate?
    • The reason as to why we focused on weight loss and not BMI change have been outlined in the above response to the reviewers second comment.
    • The rationale as to why no sample size calculation was provided has been provided as per reviewer 1 comments and the following has been added - “Due to being a service evaluation, formal sample size calculation was not conducted” [Line 210 – 211].
    • We did not conduct multi-variant analysis to assess the difference between elderly and non-elderly, due to the service evaluation only including three participants over the age of 65 years.
    • In addition we have covered several additional points from reviewer 1 which we hope have aided this section.
  • Are the methods adequately described?
    • Primary and secondary outcomes have now been outlined within the methodology to help navigate the reader and provide clarity to the results as per reviewer 1 comments - 11. Outcomes. The primary outcome was weight change at 12 months following LED programme commencement. The percentage of participants who achieve ≥5, ≥10, ≥15 and ≥20kg of weight loss was also included. Secondary outcomes reported were improvement to HbA1c and diabetes remission rates, alongside other health-related outcomes including body composition, medication changes, blood pressure, blood lipids, and mental health and eating behaviour at 12 months [Line 170–175].
  • The results are clearly presented?
    • This has been addressed as part of reviewer 1 comments regarding the presentation of the results sections as follows:
    • Figure 3A & B title has been moved above the figures [Line 292–293].
    • To add clarity to the results section, we have added subtitles to the following sections ‘highlight referral and participant results’ (1), followed by ‘clinical outcomes’ (3.2).
    • We have removed some of the within text referral data, as much of this information is provided within supplementary material. Therefore, this now reads: “Of those referred to the programme (n=216), mean age was 52.5 (SD 11.0) years, BMI 35.7 (SD 6.5) kg/m2, HbA1c 66.2 (SD 21.1) mmol/mol, 55.1% (n=119) were women, with the majority being from African (incl. Somali) ethnicity (n=41, 19%). The mean length of T2DM diagnosis was 3.7 (SD 5.2) years. Please see Supplementary Material (Table S1) for further information regarding referral data to the programme” [Line 230 – 234].
    • Additionally, we did the same to the baseline characteristic data. This now reads: "Of those participating in the programme (n=35), mean age was 50.4 (SD 10.5) years, BMI was 34.4 (SD 4.4) kg/m2, HbA1c was 65 (SD 20.3) mmol/mol and 20 (57%) were men. The majority of participants were either Black (n=7, 20%) or White British (n=7, 20%) , and the mean length of diagnosis was 4.2 (3.6) years (Table 1)” [Line 238–241].
    • By reducing the above two paragraphs, less within text data is provided, but the reader is still able to compare key data between referrals and participants.

We would like to thank the reviewer again for their time and valuable comments and hope that this has clarified the comments and points they have regarding the manuscript.
